# Essential functions of Runx/Cbfβ in gut conventional dendritic cells for priming Rorγt⁺ T cells

Mari Tenno[1],*, Alicia Yoke Wei Wong[2],*, Mika Ikegaya[1], Eiji Miyauchi[3], Wooseok Seo[1], Peter See[2], Tamotsu Kato[3], Takashi Taida[3,4], Michiko Ohno-Oishi[1], Hiroshi Ohno[3], Hideyuki Yoshida[5], Florent Ginhoux[2,6], Ichiro Taniuchi[1]

**Acquired immune responses are initiated by activation of CD4⁺ helper T (Th) cells via recognition of antigens presented by conventional dendritic cells (cDCs). DCs instruct Th-cell polarization program into specific effector Th subset, which will dictate the type of immune responses. Hence, it is important to unravel how differentiation and/or activation of DC are linked with Th-cell–intrinsic mechanism that directs differentiation toward a specific effector Th subset. Here, we show that loss of Runx/Cbfβ transcription factors complexes during DC development leads to loss of CD103⁺CD11b⁺ cDC2s and alters characteristics of CD103⁻CD11b⁺ cDCs in the intestine, which was accompanied with impaired differentiation of Rorγt⁺ Th17 cells and type 3 Rorγt⁺ regulatory T cells. We also show that a Runx-binding enhancer in the *Rorc* gene is essential for T cells to integrate cDC-derived signals to induce Rorγt expression. These findings reveal that Runx/Cbfβ complexes play crucial and complementary roles in cDCs and Th cells to shape converging type 3 immune responses.**

## Introduction

Conventional dendritic cells (cDCs) are specialized antigen-presenting cells of the immune system. DCs in the intestine lamina propria (ILP) sense diverse antigens and migrates to draining lymph nodes where they instruct CD4⁺ T helper (Th) cells to differentiate into several types of effector Th cells, such as Rorγt⁺ Th17 and Foxp3⁺ peripherally induced regulatory T (iTreg) cells (Durai & Murphy, 2016; Honda & Littman, 2016). Gut cDCs are composed of two main subsets named cDC1 and cDC2 (Guilliams et al, 2014), with specialized polarizing Th functions. Gut CD103⁺ DCs were initially reported to induce FoxP3⁺ Treg cells (Coombes et al, 2007; Sun et al, 2007). However, gut CD103⁺ DCs are now subdivided into CD103⁺CD11b⁺

cDC2 and CD103⁺CD11b⁻ cDC1. Although the functions of CD103⁺CD11b⁺ cDC2 are not fully understood, previous studies have suggested that CD103⁺CD11b⁺ cDC2 have the capacity to induce both Th17 cells (Lewis et al, 2011; Persson et al, 2013; Schlitzer et al, 2013) and iTreg cells (Bain et al, 2017). On the other hand, Foxp3⁺ iTreg cells can be divided into Rorγt⁻Foxp3⁺ iTreg and Rorγt⁺ Foxp3⁺ Treg, the latter is designated as type 3 Treg (Park & Eberl, 2018). Although the exact roles of Rorγt⁺ type 3 Treg cells have not yet been unraveled, they are involved in suppressing exaggerated Th2 responses (Ohnmacht et al, 2015), Th17 and Th1 responses (Sefik et al, 2015). However, it remains elusive which cDC subset(s) regulates the differentiation of Rorγt⁺ Th17 and Rorγt⁺ Foxp3⁺ Treg cells and how T cells integrate signals from cDCs to activate *Rorc* gene to induce Rorγt expression.

Runx transcription factor family proteins function as heterodimers with Cbfβ and regulate many types of hematopoietic cells (de Bruijn & Speck, 2004; Ebihara et al, 2017). Among three mammal Runx proteins Runx1, Runx2, and Runx3, loss of Runx3 in hematopoietic cells leads to spontaneous development of colitis (Brenner et al, 2004) and airway infiltration in part by altering DCs function (Fainaru et al, 2004). In this study, we show that Runx/Cbfβ functions in DCs are essential not only for the differentiation of intestinal CD103⁺CD11b⁺ cDC2 but also for the priming of Rorγt-expressing T cells to maintain gut homeostasis.

## Results

### Runx/Cbfβ complexes are essential for the differentiation of gut CD103⁺CD11b⁺ cDC2s

Runx/Cbfβ complexes regulate differentiation of Langerhans cells, epidermal-specific antigen-presenting cells, at least by transmitting TGFβ receptor signaling (Tenno et al, 2017). During DC differentiation in the gut, TGFβ receptor signaling was shown to be

[1]Laboratory for Transcriptional Regulation, RIKEN Center for Integrative Medical Sciences, Yokohama, Japan   [2]Singapore Immunology Network (SIgN), A*STAR, Biomedical Grove, Singapore   [3]Laboratory for Intestinal Ecosystem, RIKEN Center for Integrative Medical Sciences, Yokohama, Japan   [4]Department of Gastroenterology, Graduate School of Medicine, Chiba University, Chuo-ku, Japan   [5] Young Chief Investigators Laboratory for Immunological Transcriptomics, RIKEN Center for Integrative Medical Sciences, Yokohama, Japan   [6]Shanghai Institute of Immunology, Shanghai Jiao Tong University School of Medicine, Shanghai, China

Correspondence: ichiro.taniuchi@riken.jp
Mari Tenno's present address is Department of Immune Regulation, Research Institute, National Center for Global Health and Medicine, Ichikawa, Japan
*Mari Tenno and Alicia Yoke Wei Wong contributed equally to this work

essential for the differentiation of CD103+CD11b+ cDC2s (Bain et al, 2017). We thereby addressed the roles of Runx/Cbfβ complexes by inactivating the *Cbfb* gene during DC development using a *CD11c-Cre* transgene (*Cbfb*^F/F^: *CD11c-Cre* mice). We defined gut cDCs as CD45+CD64−CD11c+MHC-II+ cells and examined CD103 and CD11b expression. Although the differentiation of CD103+CD11b− cDC1s was not affected by loss of Cbfβ, percentage and absolute cell numbers of CD103+CD11b+ cDC2s were dramatically decreased in the small intestine, which was accompanied with increased relative numbers of CD103−CD11b+ DCs (Fig 1A). In the mesenteric lymph nodes, migratory gut DCs were defined as CD45+MHC-II^hi^CD11c^lo^ cells. As we observed in the small intestine, CD103+CD11b+ cDC2s in the migratory DC fraction were decreased in both relative and absolute cell numbers upon loss of Cbfβ (Fig 1B). CD103+CD11b+ cDC2s also tended to be decreased also in the large intestine of *Cbfb*^F/F^: *CD11c-Cre* mice (Fig S1A).

CD101 is another marker that is predominantly expressed on CD103+CD11b+ cDC2s (Bain et al, 2017), and we confirmed this observation (Fig S2A). In the small intestine of *Cbfb*^F/F^: *CD11c-Cre* mice, CD101+CD11b+ cDCs were also decreased and remaining CD101+CD11b+ cDCs were CD103−CD11b+ cells, whereas most CD101+CD11b+ cells in the control mice expressed CD103 (Fig S2A). Along with no accumulation of CD101+ cells in CD103−CD11b+ population in *Cbfb*^F/F^: *CD11c-Cre* mice (Fig S2A), we concluded that the development of CD103+CD11b+ cDC2s was severely impaired in the absence of Cbfβ.

We also examined cDCs in another barrier tissue, the lung. In lung cDC population, our analyses of CD103 and CD11b expression detected two major cDC subsets, CD103+CD11b− and CD103−CD11b+, with a very minor population of CD103+CD11b+ cells, which is consistent with a previous report (Greter et al, 2012) proposing that CD103−CD11b+ subset is a major cDC2 subset in the lung. Interestingly, absolute cell numbers of CD103−CD11b+ cDC2 subset were significantly reduced in the lung of *Cbfb*^F/F^: *CD11c-Cre* mice (Fig S2B). In addition, CD101 expression was nearly undetected in both CD103+CD11b− and CD103−CD11b+ subsets upon loss of Cbfβ, indicating that Cbfβ deficiency also resulted in the impaired differentiation of CD101-expressing CD11b+ cDC2 subset in the lung.

All three mammal *Runx* genes, *Runx1*, *Runx2*, and *Runx3*, are expressed in all cDC subsets in the small intestine (Fig S3A). Using an anti-Runx3 antibody that detects Runx3 protein in flow cytometry analyses, we found that CD103+CD11b+ and CD103−CD11b+ gut DC subsets expressed Runx3 at higher level than that in CD103+CD11b− DCs (Fig S3B). To examine which Runx protein is responsible for the development of CD103+CD11b+ cDC2s, we inactivated *Runx1* and *Runx3* gene by *CD11c-Cre* individually or in a combinational manner. Although ablation of *Runx1* or *Runx3* alone caused a slight decrease in CD103+CD11b+ cDC2s, these cells were nearly absent in the small intestine and mesenteric lymph nodes upon loss of both Runx1 and Runx3 proteins (Fig 2). Decrease in CD103+CD11b+ cDC2s was also observed in the large intestine by combination deletion of both Runx1 and Runx3 (Fig S3C), as was

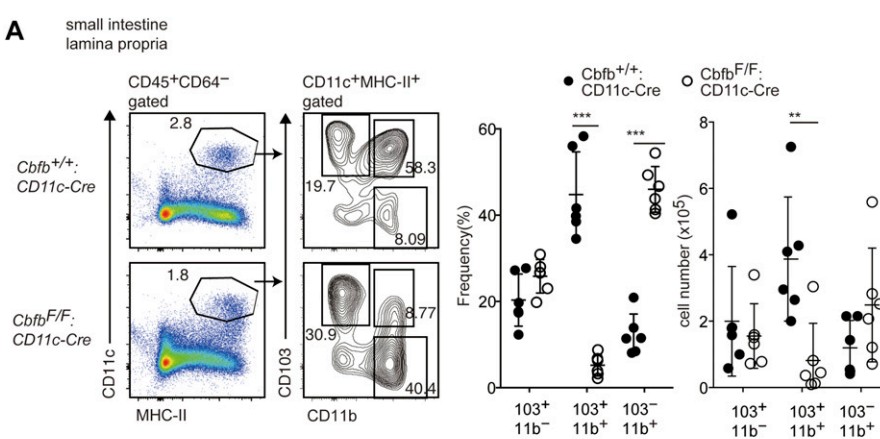

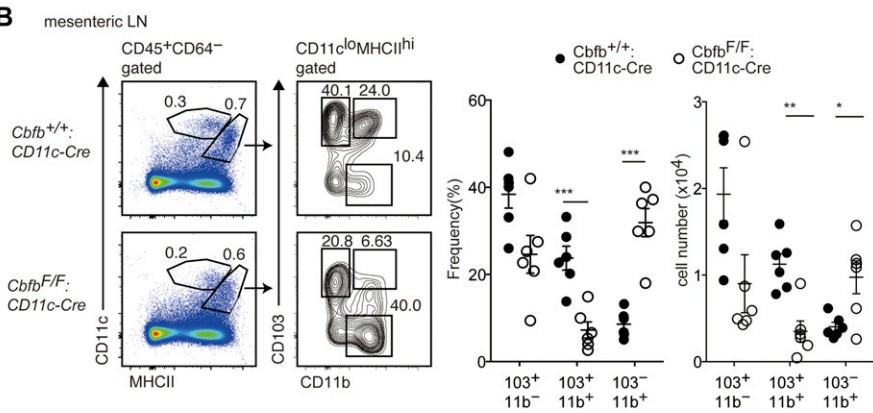

**Figure 1. Loss of CD103+CD11b+ gut DC subset in the absence of Runx/Cbfβ complexes.**
**(A)** Pseudocolor blots showing gating strategy to define small intestine DCs. Contour plots showing CD103 and CD11b expression in DCs of *Cbfb*^+/+^: *CD11c-Cre* and *Cbfb*^F/F^: *CD11c-Cre* mice. Graphs in the right show the summary of the percentage and cell numbers of indicated DCs subsets. Each dot represent individual mouse. Mean ± SD. **(B)** Pseudocolor blots showing gating strategy to define migratory DCs in mesenteric lymph nodes. Contour plots showing CD103 and CD11b expression in CD11c^lo^MHC-II^hi^ migratory DCs. Graphs in the right show the summary of the percentage and cell numbers of indicated mesenteric LN DCs subset. Each dot represent individual mouse. Mean ± SD. *P < 0.05, **P < 0.01, ***P < 0.001. Statistical analyses were performed by Student's unpaired *t* test.

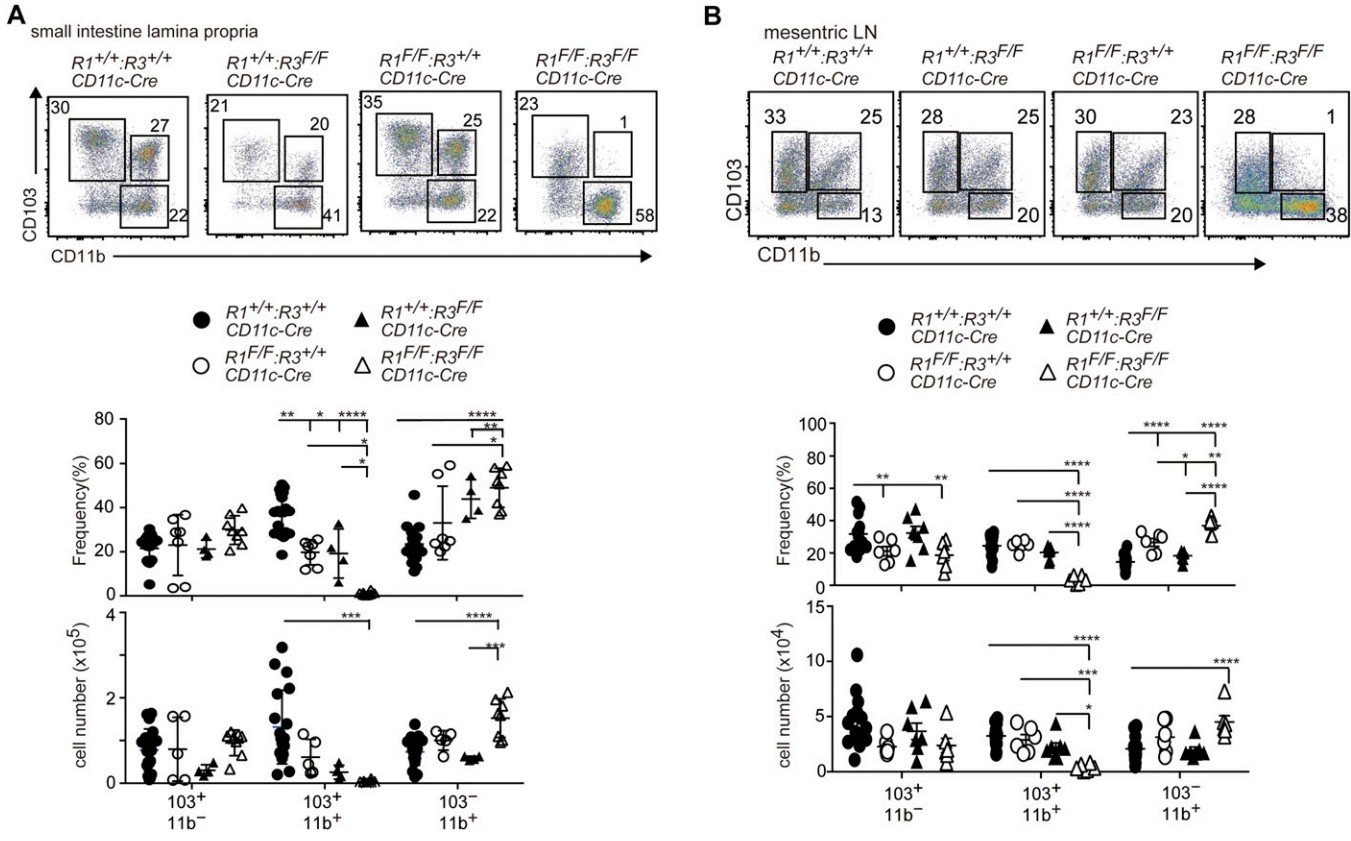

**Figure 2. Loss of CD103⁺CD11b⁺ gut DC subset in the absence of both Runx1 and Runx3.**
**(A)** Pseudocolor blots showing CD103 and CD11b expression in small intestine DCs of *Runx1(R1)⁺/⁺: Runx3(R3)⁺/⁺, R1⁺/⁺: R3^{F/F}, R1^{F/F}: R3⁺/⁺*, and *R1^{F/F}: R3^{F/F}* mice harboring *CD11c-Cre*. Graphs at the bottom show the summary of the percentage and cell numbers of indicated DCs subsets. **(B)** Pseudocolor blots showing gating strategy to define migratory DCs in mesenteric lymph nodes. Contour plots showing CD103 and CD11b expression in CD11c^{lo}MHC-II^{hi} migratory DCs. Graphs at the bottom show the summary of the percentage and cell numbers of indicated mesenteric LN DCs subset. Each dot represents individual mouse. Mean ± SD. Numbers in the plots indicate the percentage of cells in each region. *P < 0.05, **P < 0.01, ***P < 0.001. Statistical analyses were performed by Student's unpaired *t* test.

observed in *Cbfb^{F/F}: CD11c-Cre* mice. These observations indicate not only the redundant function between Runx1 and Runx3 proteins but also the essential requirement of both proteins to direct the differentiation of CD103⁺CD11b⁺ cDC2s.

### Impaired Rorγt⁺ T-cell differentiation by lack of Runx/Cbfβ in DCs

One of the important functions of gut DCs is to prime CD4⁺ Th cells to differentiate into effector Th subsets such as FoxP3⁺ iTreg and Rorγt⁺ Th17 cells (Durai & Murphy, 2016). Because CD103⁺CD11b⁺ cDC2s were shown to support differentiation of both iTreg and Th17 cells (Lewis et al, 2011; Persson et al, 2013; Bain et al, 2017), we examined the expression of FoxP3 and Rorγt in the lamina propria CD4⁺ T-cell population. In *Cbfb^{F/F}: CD11c-Cre* mice, the relative numbers of both FoxP3⁺Rorγt⁺ and FoxP3⁻Rorγt⁺ cells were significantly decreased in the small intestine (Fig 3A), whereas the reduction of such Rorγt⁺ T cells was not obvious in the large intestine (data not shown). Inactivation of both *Runx1* and *Runx3* genes in DCs also led to the decrease in small intestine Rorγt⁺ CD4⁺ T cells (Fig S3D). These observations suggest an essential role of Runx/Cbfβ in DCs to support the differentiation of Rorγt⁺ T cells.

Functions of Runx/Cbfβ in T cells were shown to be important for the development of Treg (Kitoh et al, 2009; Rudra et al, 2009) and

Th17 cells (Zhang et al, 2008). Previous report detected a leaky expression of the *CD11c-Cre* transgene in T lymphocytes (Bain et al, 2017). Similarly, we detected *CD11c-Cre*–mediated recombination of *Cbfb* gene to some extents in gut CD4⁺ T cells (Fig S1B). To examine cell intrinsic effects of *Cbfb* inactivation on the differentiation of gut Th17 and Treg cells, we inactivated Cbfb gene in T cells using CD4-Cre transgene (*Cbfb^{F/F}: CD4-Cre* mice) and compared gut CD4⁺ T-cell subsets between *Cbfb^{F/F}: CD11c-Cre* and *Cbfb^{F/F}: CD4-Cre* mice. As previously reported (Kitoh et al, 2009; Rudra et al, 2009), FoxP3 expression level was slightly decreased upon inactivation of *Cbfb* in T cells by *CD4-Cre* transgene, whereas the relative numbers of FoxP3⁻Rorγt⁺ Th17 cells were not affected (Fig 3B). Instead, loss of Cbfβ in T cells resulted in an increase in FoxP3⁻Gata3⁺ Th2 cells in the small intestine (Fig 3B), presumably in part due to a failure of *Il-4* silencing (Naoe et al, 2007) and loss of Runx3-mediated antagonism against Gata3 function (Yagi et al, 2010). We also examined the effect of combinational loss of Cbfβ in T cells with Cbfβ deficiency in DCs. In the gut of *Cbfb^{F/F}: CD11c-Cre: CD4-Cre* mice, reduction of Rorγt⁺ cells, in particular FoxP3⁺Rorγt⁺ cells, tended to be enhanced, indicating that the loss of Runx/Cbfβ in both T cells and DC synergistically impacts the differentiation of Rorγt⁺ T cells.

We previously showed that Rorγt⁺ expression in group 3 innate lymphoid cells (ILC3) and lymphoid tissue inducer (LTi) cells is lost by

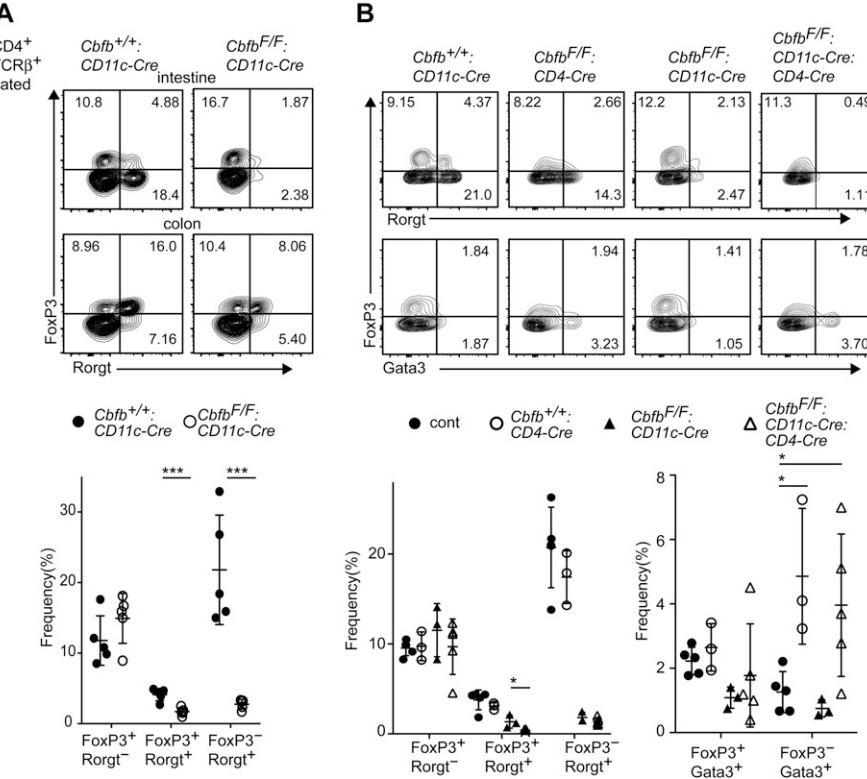

**Figure 3. Impaired differentiation of Rorγt⁺ Treg and Th17 cells by loss of Runx/Cbfβ complexes in DCs.**
**(A)** Contour plots showing FoxP3 and Rorγt expression in small intestine CD4⁺TCRβ⁺ cells of *Cbfb⁺/⁺: CD11c-Cre* and *Cbfb^F/F: CD11c-Cre* mice. Graph at the bottom shows the summary of the percentage of indicated T-cell subsets. **(B)** Contour plots showing FoxP3, Rorγt and Gata3 expression in small intestine CD4⁺TCRβ⁺ cells of *Cbfb⁺/⁺: CD11c-Cre*, *Cbfb^F/F: CD4-Cre Cbfb^F/F: CD11c-Cre*, and *Cbfb^F/F: CD11c-Cre: CD4-Cre* mice. Graphs at the bottom show the summary of the percentage of indicated T-cell subsets. Numbers in the contour plots indicate the percentage of cells in each quadrant. Each dot represents individual mouse. Mean ± SD. *P < 0.05, ***P < 0.001. Statistical analyses were performed by Student's unpaired *t* test.

removal of a single intronic enhancer (*E-11*), which locates 11 kb downstream from the *Rorgt* transcriptional start site and is occupied by Runx/Cbfβ complexes in thymocytes (Tenno et al, 2018). To further examine the Runx-mediated T-cell–intrinsic mechanism for Rorγt⁺ induction, we first addressed a role of the *E-11* for Rorγt expression in T cells using the *Rorc^gfp:ΔE-11* reporter allele in which the *E-11* enhancer was removed from the *Rorc^gfp* allele. In the small ILP of *Rorc⁺/gfp:ΔE-11* mice, Rorγt⁺ T cells were developed. However, GFP expression from the *Rorc^gfp:ΔE-11* reporter allele was not detected (Fig S4A). Similarly, Rorγt⁺ cells were absent from the small ILP of *Rorc^ΔE-11/ΔE-11* mice lacking the *E-11* enhancer in the *Rorc* locus. In addition, Rorγt-expressing cells in non–T-cell population, which correspond to ILC3 and LTi, were lost in *Rorc^ΔE-11/ΔE-11* mice (Fig S4B), whereas the level of Rorγt⁺ in DP thymocytes was just reduced to the half of the one detected in control cells. In addition, naïve CD4⁺ T cells from *Rorc^ΔE-11/ΔE-11* mice failed to differentiate into IL-17–producing Th17 cells in Th17-inducing culture condition (Fig S4B). These results demonstrated that the *E-11* enhancer is crucial for Rorγt⁺ expression during in vivo and in vitro differentiation to Th17, whereas other regulatory regions partially compensate Rorγt induction in DP thymocytes.

## Impaired signatures of CD103⁻CD11b⁺ cDC2 in the absence of Cbfβ

Our results showed that the loss of Runx/Cbfβ during cDC development resulted in loss of CD103⁺CD11b⁺ cDC2s and a severe reduction in Rorγt-expressing T cells in the small intestine. A specific loss of CD103⁺CD11b⁺ cDC2s was also observed in other mutant mouse strains, such as the *Notch2^F/F: CD11c-Cre*, *Irf4^F/F: CD11c-Cre*, and *Tgfbr2^F/F: CD11c-Cre* mice (Lewis et al, 2011; Persson et al, 2013;

Satpathy et al, 2013; Schlitzer et al, 2013; Bain et al, 2017). However, the degree of Th17 cells reduction reported in those mice seems to be less than that in *Cbfb^F/F: CD11c-Cre* mice. Indeed, a significant proportion of Rorγt⁺ T cells remained in *Notch2^F/F: CD11c-Cre* mice (Fig 4A), although the relative numbers of Rorγt⁺ T cells tended to be reduced when compared with that of control mice. Thus, the relative numbers of Rorγt⁺ T cells were more than twofolds lower in *Cbfb^F/F: CD11c-Cre* mice than that in *Notch2^F/F: CD11c-Cre* mice (Figs 3A and 4A), whereas the degree of CD103⁺CD11b⁺ cDC2s reduction was comparable between these two strains. Interestingly, the percentages of CD103⁺ cells in CD101⁺CD11b⁺ cDC2s of *Notch2^F/F: CD11c-Cre* mice were lower than that of control mice but were higher than that of *Cbfb^F/F: CD11c-Cre* (Fig 4B). These observations suggested that the impaired survival of CD103⁺CD11b⁺ cDC2s might occur by loss of Notch2, whereas developmental inhibition is likely to be the main mechanism for reduction of CD103⁺CD11b⁺ cDC2s in *Cbfb^F/F: CD11c-Cre* mice, as was observed in the case of the loss of TGFβ receptor signaling (Bain et al, 2017).

It is conceivable that the severer loss of Rorγt⁺ T cells in *Cbfb^F/F: CD11c-Cre* is not caused simply by the loss of CD103⁺CD11b⁺ cDC2s, and that the functions of other DCs supporting Rorγt⁺ T-cell differentiation might also be impaired by the loss of Cbfβ. We, therefore, examined the gene expression profiles of Cbfβ-deficient CD103⁺CD11b⁻ and CD103⁻CD11b⁺ DCs by RNA-seq. Principle component analyses (PCA) indicated that CD103⁻CD11b⁺ DCs became more different from control cells in PC2 values upon loss of Cbfβ (Fig 5A). In CD103⁻CD11b⁺ DCs, more genes were up-regulated upon loss of Cbfβ, such as IL-18, than down-regulated in their expression levels (Fig 5B and C). Interestingly, IL-18 was shown to limit colonic

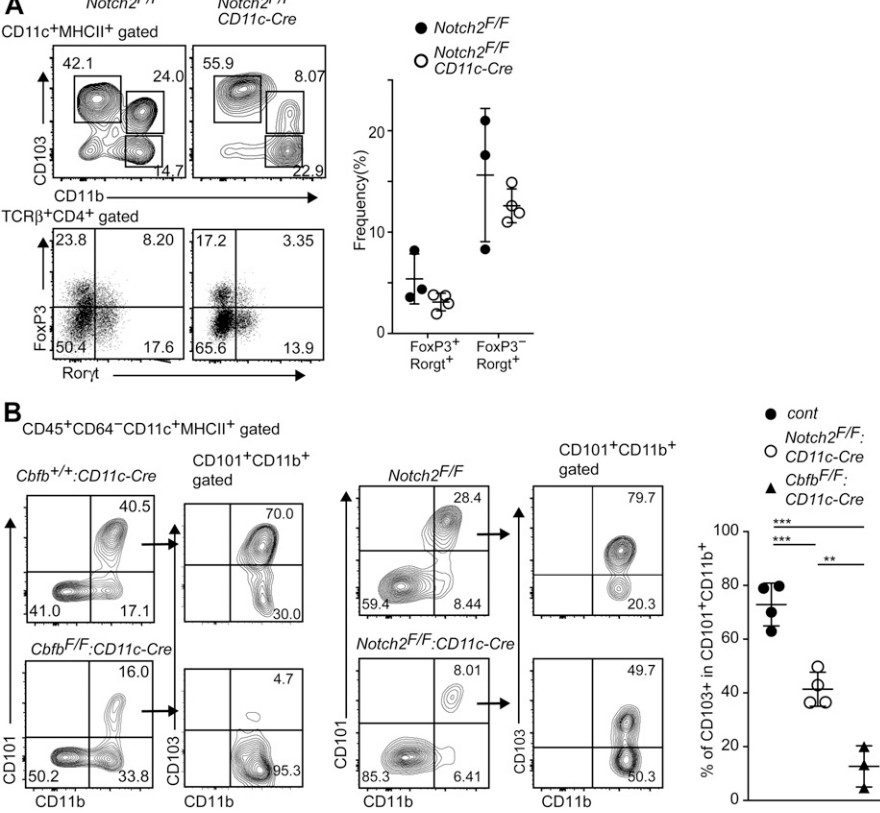

**Figure 4. Unique requirement for Runx/Cbfβ in regulating differentiation of CD103⁺CD11b⁺ cDCs.**
**(A)** Contour plots showing CD103 and CD11b expression in small intestine CD11c⁺MHC-II⁺ DCs (upper) and FoxP3 and Rorγt expression in CD4⁺TCRβ⁺ cells (lower) of *Notch2^F/F^*, *Notch2^F/F^: CD11c-Cre* mice. Graph at the right shows the summary of the percentage of indicated T-cell subsets. **(B)** Contour plots showing CD101 and CD11b expression in small intestine CD11c⁺MHC-II⁺ DCs of *Cbfb^+/+^: CD11c-Cre*, *Cbfb^F/F^: CD11c-Cre*, and *Notch2^F/F^: CD11c-Cre* mice. Expression of CD103 and CD11b in CD101⁺CD11b⁺ cells are shown in the right contour plots. Graph at the right shows the summary of the percentage of CD101⁺CD11b⁺ cells of *Cbfb^+/+^: CD11c-Cre*, *Cbfb^F/F^: CD11c-Cre*, and *Notch2^F/F^: CD11c-Cre* mice. Each dot represents individual mouse. Mean ± SD. **P < 0.01, ***P < 0.001. Statistical analyses were performed by Student's unpaired *t* test.

Th17 cell differentiation (Harrison et al, 2015). On the contrary, in CD103⁺CD11b⁻ DCs, more genes were down-regulated upon loss of Cbfβ, than up-regulated (Fig 5A). Recently, IL22ra2, one of the down-regulated genes in both CD103⁺CD11b⁻ and CD103⁻CD11b⁺ DCs, was shown to be decreased by IL-18 and be involved in the control of intestinal tissue damage (Huber et al, 2012).

### Spontaneous colitis development in *Cbfb^F/F^: CD11c-Cre* mice

Our results indicate that the loss of Cbfβ during cDC development led to significant changes in cDC and Th-cell composition in the intestine. It was reported that Runx3 loss in hematopoietic cells (Brenner et al, 2004) and loss of CD103⁺CD11b⁺ cDC2s by lack of TGFβ receptor signaling (Bain et al, 2017) resulted in spontaneous colitis development. Interestingly, *Cbfb^F/F^: CD11c-Cre* mice tended to develop colitis spontaneously around 6 mo and later, although significant histological changes were not observed at 7 wk old when differentiations of gut DCs and Th subsets were analyzed (Figs 6A and S5). Despite different cellular compositions in the intestine and colitis development in *Cbfb^F/F^: CD11c-Cre* mice, the composition of the commensal gut microbiota assessed by feces of 5-mo-old *Cbfb^F/F^: CD11c-Cre* mice were not significantly different from that in control mice (Fig S6), except for an increase in segmented filamentous bacteria (SFB) (Fig 6B), a known inducer of Th17 differentiation (Ivanov et al, 2009). These observations confirmed that the decreased Th17 differentiation that we observed is not caused by lack of SFB in microbiota.

## Discussion

In this work, we provide concrete genetic evidence supporting a crucial function of Runx/Cbfβ complexes in gut cDC development, not only for the differentiation of CD103⁺CD11b⁺ cDC2s but also for their polarizing activity that primes both Rorγt⁺ Th17 and type 3 Rorγt⁺ FoxP3⁺ Treg cells. Several mutant mouse lines, in which CD103⁺CD11b⁺ cDC2s in the gut was diminished or lost, have been reported. However, roles of gut cDCs in priming gut Th-cell subset still remains elusive. For example, loss or reduction of CD103⁺CD11b⁺ cDC2s in intestinal lamina upon inactivation of *Notch2* (Lewis et al, 2011; Satpathy et al, 2013) or *Irf4* genes (Schlitzer et al, 2013) or by the expression of the Langerin-diphtheria toxin A transgene (Welty et al, 2013) was shown to result in a decrease in Th17 cells without significant reduction in Foxp3⁺ Treg cells. In contrast, ablation of TGFβR during DC development led to the reduction in CD103⁺CD11b⁺ cDC2s, which was accompanied with reductions of both Th17 and Foxp3⁺ Treg cells (Bain et al, 2017). It is noteworthy that gut Rorγt⁺ CD4⁺ T cells consists of at least two types of cells, Th17 and type 3 Treg cells. Although staining of both FoxP3 and Rorγt proteins is necessary to detect these two subsets, previous studies did not always examine these two subsets, making it difficult to fairly compare the phenotype of gut effector Th cells among several mouse strains. However, reduction of Rorγt⁺ T cells, including Th17 and type 3 Treg, seems to be more severe in *Cbfb^F/F^: CD11c-Cre* mice than in *Tgfbr1^F/F^: CD11c-Cre* mice, suggesting that Runx/Cbfβ complexes may have unique roles other than

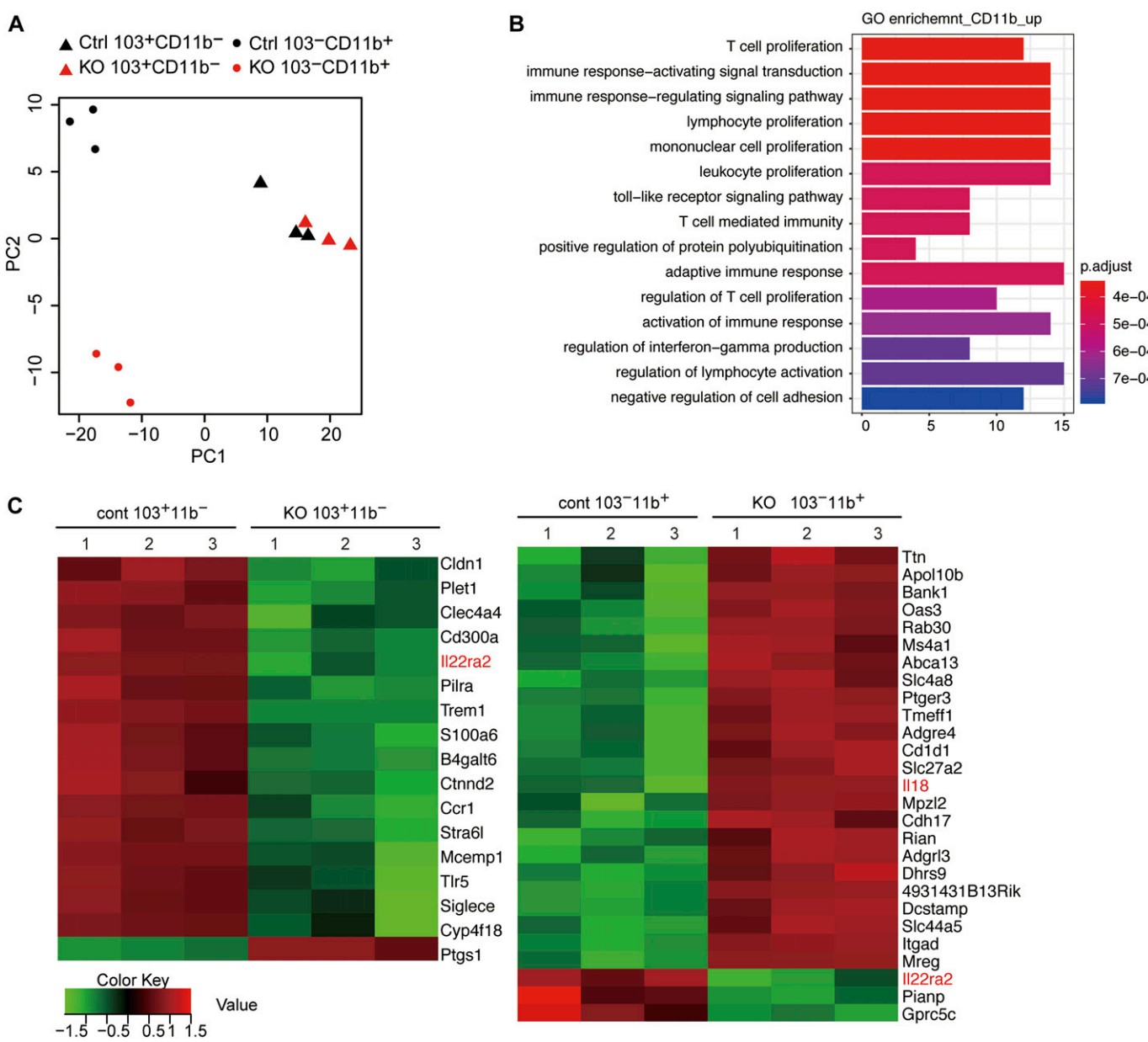

**Figure 5. Requirement for Runx/Cbfβ complexes for CD103⁻CD11b⁺ function supporting differentiation of Rorγt⁺ T cells.**
**(A)** PCA of RNA-seq showing relationship of *wild-type* (Ctrl) and *Cbfb^{F/F}: CD11c-Cre* (KO) CD103⁺CD11b⁻ (triangle) and CD103⁻CD11b⁺ (oval) small intestine DCs. **(B)** Gene Ontology enrichment analyses showing up-regulated gene signatures in *Cbfb^{F/F}: CD11c-Cre* CD103⁻CD11b⁺ cDC subset relative to control CD103⁻CD11b⁺ cDC subset. **(C)** Heat map showing most differentially expressed genes (fivefold changes, FDR < 0.01) in CD103⁺CD11b⁻ (left) and CD103⁻CD11b⁺ (right) DCs relative to control cells detected by RNA-seq. RNA-seq were performed with three samples sorted from three different mice.

transmitting TGFβ signals in regulating development or functions of gut cDCs. In line with this assumption, we found that CD101 expression in the gut and lung cDCs was nearly abolished in *Cbfb^{F/F}: CD11c-Cre* mice (Fig S2), whereas it was detected on those cells after loss of TGFβR (Bain et al, 2017). This difference suggests that Runx/Cbfβ complexes might regulate DC development during earlier stages rather than later stages, which requires TGFβ signaling. CD101 expression on T lymphocytes and myeloid cells was reported to be important for preventing intestinal inflammation (Schey et al, 2016). Although relevance of CD101 expression in regulating cDCs remains

unclear, it is possible that induction of CD101 expression during cDC differentiation is important for their polarizing functions of effector Th-cell differentiation.

It remains unclear whether a single DCs subset plays a predominant role in inducing both Th17 and type 3 Treg cells or distinct subpopulations of these CD103⁺CD11b⁺ cDC2s prime each T-cell subset independently. Compared with other mutant strains lacking CD103⁺CD11b⁺ cDC2s subset, the reduction in Rorγt-expressing T cells is prominent upon the loss of Cbfβ. Such clear difference may suggest a presence of another DC population that compensate the

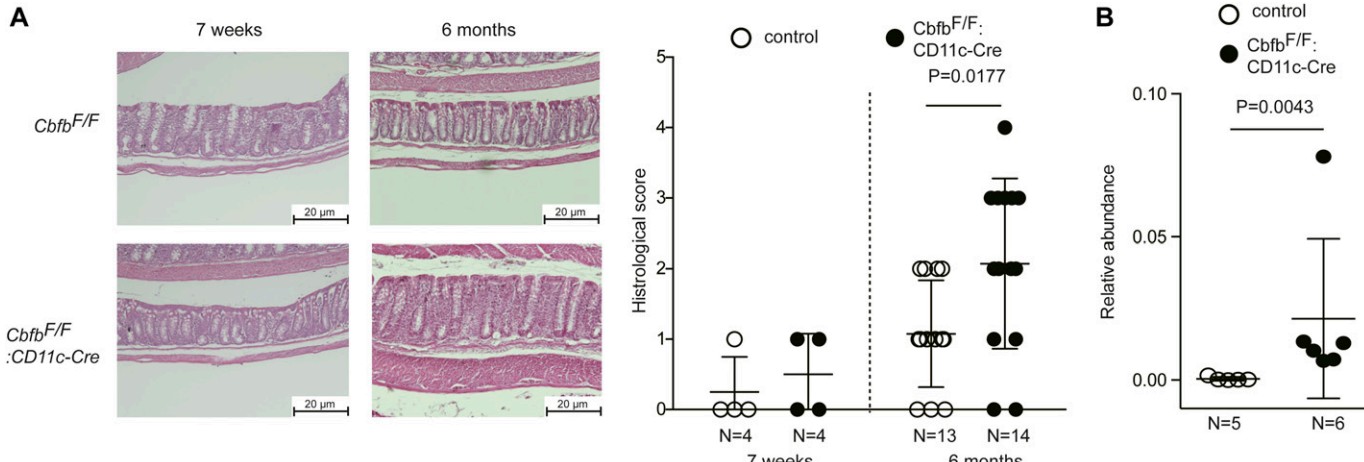

**Figure 6. Spontaneous colitis development by lack of Cbfβ in gut DCs.**
**(A)** Representative colon histology of 7-wk and 6–9-mo-old control and *Cbfb^{F/F}: CD11c-Cre* mice by hematoxylin and eosin staining. Scale bar, 20 μm. Graph at the right shows the summary of histological scoring. Scale bar, 20 μm. **(B)** Results of gut microbiotas analyses of 5-mo-old mice showing increased proportion of SFB in *Cbfb^{F/F}: CD11c-Cre* mice. P values were analyzed by Mann–Whitney test.

loss of CD103⁺CD11b⁺ cDC2 subset for inducing Rorγt⁺ T-cell differentiation. Previously, CD103⁻CD11b⁺ DCs was shown to have capacity to prime Th17 differentiation (Liang et al, 2016). Residual CD103⁻CD11b⁺ cDCs in *Cbfb^{F/F}: CD11c-Cre* mice showed impaired gene signatures including up-regulation of the *Il-18* gene. Given a reported antagonistic function of IL-18 against Th17 differentiation (Harrison et al, 2015), ectopic IL-18 secretion from Cbfβ-deficient CD103⁻CD11b⁺ cDCs might alter the microenvironment toward inhibiting Th17 differentiation and intestinal inflammation. Consistent with a previous report showing a role of IL-18 in decreasing IL22ra2 level (Huber et al, 2012), IL22ra2 was down-regulated in both CD103⁺CD11b⁻ and CD103⁻CD11b⁺ cDC subsets. Thus, it is possible that dysfunction of remaining cDCs by loss of Runx/Cbfβ have direct or indirect roles in regulating Rorγt⁺ T-cell differentiation. Either case, Runx/Cbfβ complexes are indispensable for either differentiation or function of such cDCs subset(s).

In addition to DC function, we showed that the single intronic enhancer, *E-11*, in the *Rorc* locus is crucial to integrate signals from cDCs to induce Rorγt expression in T cells. The *E-11* enhancer contains two conserved Runx-binding sequences and is occupied by Runx/Cbfβ in thymocytes (Tenno et al, 2018). Rorγt⁺ T cells are still differentiated upon loss of Cbfβ in T cells (Fig 3B), suggesting that other transcription factors compensate Runx/Cbfβ function for activating the *E-11* enhancer. However, once DC-derived priming signals are diminished by loss of Cbfβ in cDCs, the effect of Cbfβ deficiency in T cells would be beyond such compensatory pathway. On the contrary, loss of ILC3 and LTi cells by attenuation of Runx/Cbfβ activity (Tachibana et al, 2011; Ebihara et al, 2015; Tenno et al, 2018) indicates that activation of the *E-11* enhancer in ILCs is more dependent on Runx/Cbfβ complexes.

Collectively, our results demonstrate that a single transcription factor, Runx/Cbfβ, plays an essential role in shaping type 3 immune responses by regulating the developmental program of distinct immune cells, cDCs and T cells. It is unclear whether DCs and T cells coincidentally use the same transcription factor to enhance type 3 responses, or, alternatively, whether there is uncharacterized crosstalk between cDCs and T cells that modulates the type 3 developmental program in each cell type. Evolutional pressures to

efficiently mount type 3 immune responses may have led to convergence of each program, leading to the use of a common transcription factor, Runx/Cbfβ. Along with an antagonism against type 2 responses in part through suppressing IL-4 production (Djuretic et al, 2007; Naoe et al, 2007), we show here that Runx/Cbfβ complexes serve as central players for balancing type 2 and type 3 immune responses.

# Materials and Methods

## Mice

*Cbfb^{flox}* (Naoe et al, 2007), *Runx1^{flox}* (Taniuchi et al, 2002), *Runx3^{flox}* (Naoe et al, 2007), *Rorc^{gfp;ΔE−11}*, and *Rorc^{ΔE−11}* mice (Tenno et al, 2018) were previously described. *Rorc^{gfp}* reporter mice (Eberl et al, 2004), *Cd11c-Cre* (Caton et al, 2007) mice, and *Cd4-Cre* (Lee et al, 2001) mice were obtained from Dr DR Littman, Dr Boris Reis, and Dr CB Wilson, respectively. *Notch2^{flox}* mice were purchased from Jackson laboratory (010525). *Cbfb^{flox}* strains were maintained in the animal facility at the RIKEN Center for Integrative Medical Sciences. *Runx1^{flox}*, *Runx3^{flox}*, and *Notch2^{flox}* strains were maintained in the animal facility at the SIgN. All animal experiments were performed in accordance with the protocol (28-017) approved by Institutional Animal Care and Use Committees of RIKEN Yokohama Branch and those approved by Institutional Animal Care and Use Committees of the Biological Resource Center (Agency for Science, Technology and Research, Singapore) and with the guidelines of animal care in RIKEN, Center for Integrative Medical Sciences, and the guidelines of the Agri-Food and Veterinary Authority (AVA) and the National Advisory Committee for Laboratory Animal Research of Singapore.

## Cell preparation

Single-cell suspensions from mouse intestine were prepared using a previously described method (Bogunovic et al, 2009) with some

modifications. Briefly, the intestinal tissues were isolated, the intestine cut open longitudinally along the entire length, and then followed by removal of Peyer's patches and fat. After washing with PBS, the tissues were cut into small pieces (5 mm in length) and incubated in PBS with 5 mM EDTA at 37°C for 20 min with shaking. After washing with PBS, the tissues were digested with 0.2 mg/ml collagenase IV (Sigma-Aldrich) in RPMI supplemented with 2% FCS at 37°C for 30 min with shaking. To homogenize samples, we placed tissues in syringe with 18G needle and flushed them out for 10 times, and then filtered them with 70-$\mu$m cell strainers. The cells were collected by centrifugation at 300$g$ at 4°C for 5 min. The pellets were washed once with PBS and were subjected for analysis.

### Flow cytometry analyses

Single-cell suspension were stained with following antibodies purchased from BD Bioscience or eBioscience: CD3 (145-2C11), CD4 (RM4-5), CD8 (53-6.7), CD11b (M1/70), CD11c (HL3), CD45(30-F11), CD64 (X54-5/7.1), CD101(Moushi101), CD103 (2E7), TCR$\beta$ (H57-597), MHC-II (M5/114.15.2), Gata3 (TWAJ), FoxP3(FJK), and Ror$\gamma$t (B2D). Runx3 antibody was purchased from BD Bioscience (R3-5G4). For intracellular staining with antibodies, the cells were permeabilized with eBioscience Foxp3/Transcription Factor Staining Buffer Set (00-5523-00). Multicolor flow cytometry analysis was performed using a BD FACSCanto II (BD Bioscience), and data were analyzed using FlowJo (Tree Star) software. Cell subsets were sorted using a BD FACSAria II (BD Biosciences).

### In vitro Th17 differentiation

Naïve CD4$^+$ T cells were purified from spleens by depleting CD4-negative cells, followed by enrichment of CD62L$^+$ cells using Naïve CD4$^+$ T cell Isolation Kit (Miltenyi Biotec) according to the manufacture's protocol. $1.0 \times 10^5$ cells were primed with plate-bound 2 $\mu$g/ml anti-CD3 (145-2C11; BD Pharmingen) and 2 $\mu$g/ml anti-CD28 (37.51; BD Pharmingen) in 100 $\mu$l DMEM supplemented with 10% FCS in round-bottomed 96-well plates at 37°C for 3 d in the presence of 10 $\mu$g/ml anti-IL-4 and anti-IFN-$\gamma$ neutralizing antibodies (BioLegend), 50 ng/ml mIL-6 (R&D) and 1 ng/ml mouse Transforming Growth Factor-$\beta$ (R&D). At day 3, 100 $\mu$l of fresh medium with the above reagents were added. At day 5, the cells were stimulated with PMA/ionomycin and fixed and permeabilized with the Fixation/Permeabilization Solution kit (BD Biosciences) and were stained for IL-17 (TC11-18H10.1, BioLegend) according to the manufacture's protocol.

### RNA-seq analysis

Expression level of Runx genes in DCs subset in wild-type mice were analyzed using GSE100393 data. To compare gene expression profiles of wild-type and Cbf$\beta$-deficient DCs subset, RNA-seq libraries were prepared by applying a RamDA-seq method for single cell RNA-seq (Hayashi et al, 2018) with some modifications. Briefly, 218~1000 CD103$^+$CD11b$^-$ and CD103$^-$CD11b$^+$ cells were sorted by FACSAria directly into tubes containing buffer TCL (QIAGEN). RNA was isolated using Agencourt RNAClean XP beads (Beckman Coulter). Then the RNA was eluted into 10 $\mu$l of RNase-free water

and used for double strand (ds)-cDNA synthesis by RamDA-seq protocol adjusted for bulk input, which used 10 times larger reaction scale than for a single cell. After quantitation of ds-cDNA using Qubit dsDNA HS Assay Kit (Thermo Fisher Scientific), 3 ng of ds-cDNA was applied for Tn5 tagmentation in a 75-$\mu$l reaction with in-house Tn5 enzyme as previously described (Hennig et al, 2018). Then the DNA was amplified by 16~19 cycles of PCR using indexed primers and KAPA HiFi DNA Polymerase (Kapa Biosystems). The libraries were purified by Agencourt AMPure XP beads (×1.2 vol.; Beckman Coulter), quantified by quantitative PCR and pooled, and were sequenced on an Illumina HiSeq 2500 instrument in a high-output mode. Low-quality part of short reads and adapter sequences were trimmed by trim_galore (v0.5.0, http://www.bioinformatics.babraham.ac.uk/projects/trim_galore/) and mapped to mm10 whole genome using STAR v2.7 with a GTF file from EMBL-EBI (GRCm38.92). Unmapped reads and reads mapped with low-quality scores (MAPQ < 5, 12~20%) and duplicated reads (14~39%) were filtered out using SAMtools view -q 5 (Li et al, 2009) and Picard Tools (Picard MarkDuplicates, http://broadinstitute.github.io/picard). Reads on each gene were counted using htseq-count (Anders et al, 2015) with -s no option and a GTF file from EMBL-EBI (GRCm38.92). To avoid non-informative genes, we removed genes on which counts were less than five in any samples from downstream analysis. Differentially expressed genes (2,560 genes, FDR < 0.01) were extracted with the "edgeR" package in R programming language (Robinson et al, 2010) to perform PCA using the "prcomp" function in R. To examine the most affected genes by Cbf$\beta$ deficiency in CD103$^+$CD11b$^-$ DCs and CD103$^-$CD11b$^+$ DCs, genes were further selected by fold change (fold change > 5, control versus Cbf$\beta$-deficient cells) and were plotted as heat map after z-score normalizations. For Gene Ontology enrichment analysis, 258 genes up-regulated in Cbf$\beta$-deficient CD103$^-$CD11b$^+$ DCs (fold change > 1.3 and FDR < 0.05, control versus Cbf$\beta$-deficient cells) were investigated using the "clusterProfiler" package in R (v3.8.1) (Yu et al, 2012).

### Histology

Colonic tissues were processed as Swiss rolls (Bialkowska et al, 2016), fixed in Mildform 10N (Wako) overnight, embedded in paraffin, and sectioned at 5-$\mu$m thickness. Histological analysis was performed after staining with hematoxylin and eosin, and the severity of colitis was scored based on the criteria (Table S1) that was previously described (Asseman et al, 1999) in a blinded fashion.

### Gut microbiotas analyses

Feces were collected from 20 to 30-wk-old mice. Bacterial DNA for 16S rRNA gene sequencing was extracted as previously described (Kato et al, 2018). The V4 variable region (515F to 806R) of the respective samples was sequenced using Illumina MiSeq by following the method previously described (Kozich et al, 2013). Taxonomic assignments and the estimation of relative abundances from sequencing data were performed using the analysis pipeline of the QIIME software package (Caporaso et al, 2010). An operational taxonomic unit (OTU) was defined at 97% similarity. OTU taxonomy was assigned based on comparison with the Greengenes database,

using UCLUST (Quast et al, 2013). Beta diversity was calculated using the UniFrac distance metric and visualized by principal coordinate analysis (PCoA) using weighted or unweighted UniFrac distances based on the OTU distribution across samples.

### Genomic PCR analyses

To examine the genotype of ILP CD4[+] T cells and MHC-II[+] CD11c[+] DCs of *Cbfb[F/F]: CD11c-Cre* mice, DNA was prepared from sorted 5000 cells and analyzed by PCR. Primers used are CbfbEx5G2 5′-CCTCCTCATTCTAA-CAGGAAT-3′, CbfbEx5G3 5′-GGTTAGGAGTCATTGTGATCAC-3′ and CbfbEx5G6 5′-CATTGGATTGGCGTTACTGG-3′.

### Data analyses

Statistical analysis was performed by Student's unpaired *t* test with GraphPad Prism6 (GraphPad software) unless otherwise stated.

## Data Availability

The sequencing data from this publication have been deposited to the Genome Expression Omnibus database under accession number GSE130380.

## Supplementary Information

## Acknowledgements

We thank N Yoza for cell sorting, M Kawasumi for technical help in histological analyses, and Y Taniguchi for mouse genotyping. This work was supported by Grants-in-Aid for Scientific Research (B) (19390118) from Japan Society for the Promotion of Science, the Grants-in-Aid for Scientific Research on Innovative Areas (17H05805) from the Ministry of Education, Culture, Sports, Science and Technology in Japan (I Taniuchi), by RIKEN Center for Integrative Medical Sciences Young Chief Investigators program (H Yoshida), and by an European Molecular Biology Organization, Young Investigator Programme, Singapore Immunology Network (SIgN) core funding, a Singapore National Research Foundation Senior Investigatorship (NRFI) NRF2016NRF-NRFI001-02 (F Ginhoux).

### Author Contributions

M Tenno: formal analysis and investigation.
AYW Wong: formal analysis and investigation.
M Ikegaya: formal analysis and investigation.
E Miyauchi: formal analysis and investigation.
W Seo: formal analysis.
P See: formal analysis.
T Kato: formal analysis.
T Taida: formal analysis.
M Ohno-Oishi: formal analysis.
H Ohno: formal analysis and supervision.
H Yoshida: software, formal analysis, and writing—review and editing.
F Ginhoux: supervision, funding acquisition, and writing—original draft, review, and editing.
I Taniuchi: supervision, funding acquisition, and writing—original draft, review, and editing.

### Conflict of Interest Statement

The authors declare that they have no conflict of interest.

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
