## [Reviewer comments · Life Science Alliance]

Life Science Alliance

Essential functions of Runx/Cbfb in gut conventional dendritic cells for priming Rorgt⁺ T cells.

Mari Tenno, Alicia Wong, Mika Ikegaya, Eiji Miyauchi, Wooseok Seo, Peter See, Tamotsu Kato, Takashi Taida, Michiko Ohno-Oishi, Hiroshi Ohno, Hldeyuki Yoshida, Florent Ginhoux, and Ichiro Taniuchi

DOI: <https://doi.org/10.26508/lsa.201900441>

Corresponding author(s): Ichiro Taniuchi, RIKEN, Center for Integrative Medical Sciences (IMS)

Review Timeline:

Submission Date:	2019-05-27
Editorial Decision:	2019-07-10
Revision Received:	2019-10-31
Editorial Decision:	2019-11-21
Revision Received:	2019-11-26
Accepted:	2019-11-27

Scientific Editor: Andrea Leibfried

Transaction Report:

July 10, 2019

Re: Life Science Alliance manuscript #LSA-2019-00441-T

Ichiro Taniuchi
RIKEN, Center for Integrative Medical Sciences (IMS)
1-7-22, Suehiro-cho, Turumi-ku
Yokohama, Kanagawa 230-0045
Japan

Dear Dr. Taniuchi,

Thank you for submitting your manuscript entitled "Essential functions of Runx/Cbfb in gut conventional dendritic cells for priming Rorgt+ T cells." to Life Science Alliance. The manuscript was assessed by expert reviewers, whose comments are appended to this letter.

As you will see, the reviewers appreciate the genetic approach employed, but think that cause and consequence leading to reduced Rorgt expression in T cells remains rather unclear, and that the Cre line used may have off target activity in T cells, not allowing to draw definitive conclusions. Furthermore, the concern was raised that the Cbfb:CD11b-Cre mice may develop colitis much earlier, leading to inflammation and therefore potentially affecting the results.

I think that some of the other concerns raised by the reviewers can be addressed by text changes to clarify, provide more context, and to avoid overstatements. The concern regarding the specificity of the Cre line (rev#2, point 3), colitis (rev#3, point 4), differences to prior work (rev#2, point 1) and the missing controls for the Runx factor analyses (rev#3, point 1 and 2) would however need to get addressed in a really good way. Should you be able to address these concerns, I'd be happy to consider a revised version for publication here.

I would be happy to discuss the individual revision points further with you should this be helpful.

I hope that the comments below will prove constructive as your work progresses.

Thank you for this interesting contribution to Life Science Alliance. I am looking forward to receiving your revised manuscript.

Sincerely,

B. MANUSCRIPT ORGANIZATION AND FORMATTING:

Reviewer #1 (Comments to the Authors (Required)):

I am in favor of publication after minor changes to the discussion.

I agree with the summary provided by the authors at the end of the presentation of the results, which states "we provide concrete genetic evidence supporting a crucial function of Runx/Cbfbeta complexes in gut DC development not only for ... CD103+CD11b+ cDC2, but also for their polarizing activity that prime both Ror γ t+ Th17 and type 3 Ror γ t FoxP3+ Treg cells".

This is a fair summary. And this summary is actually about all the paper really says. There are limitations in the study that are implicit in this statement. The evidence provided in the data is limited exclusively to genetic evidence, and there is absolutely no mechanistic data provided at any other level to explain the how loss of Cbfb in DCs causes a reduction in Ror γ t expression in T cells. The genetic evidence applies only to the requirement for Runx/Cbfbeta in DCs. This data does not indicate that the decrease in Ror γ t expression in T cells was even caused by the absence of the CD103+CD11b+ cDC2. Yes, that subset of DC2 is missing, but Runx/Cbfbeta complexes are missing from all types of other DCs that remain in the mouse. The genetic evidence does not indicate if the impact on Ror γ t in T cells is due to loss of a DC population or from loss of function in the DCs that remain.

Since the authors provide no functional mechanism for how the DCs of any type impact the development of the Ror γ t+ T cells, the authors need to be very careful to not over extend their conclusions. They run the risk of adding to a dogma that CD103+ CD11b+ DC2 drive Ror γ t in the T cells. No such evidence is there. This area in the DC field already has conflicting data as to these DC2 in the gut, with respect to various models, the deletion of Notch2 by CD11c-Cre compared to deletion of these cells by a transgene DT reagent generated by Kaplan and used by several groups. The main point is that there is no underlying mechanism for how any DC2 subset exerts a skewing on T cells (whether it is Notch2, Runx, KLF4). It is just not known at this time. The expression arrays shown in Figure 4 do not answer it.

Regarding publication, I think that this journal is a fine venue to publish the data, as is, and I have no problems with any of the figures or experiments. I don't really like the inclusion of the E-11 enhancer data in this paper myself, but I can understand the need at times to combine orphaned data, and that is what this seems like to me. It adds a figure or two, but it really adds nothing related to the mechanism of the DC effect on T cells, in the end. But keep it in, and discuss separately.

What I whole heartedly encourage the authors to do is to re-write much of the latter part and emphasize the parts of the story that still are totally unclear. By that I mean missing cells does not implicate them as the cause of the effect, as the Runx/Cbfb is gone in the other DCs as well, and some reviewer's would even complain about the CD11c-Cre, since some deletion in other cells might occur, but I will not complain. The authors have under-cited some papers, and this journal's citation style is really annoying, I must say. The authors should also expose the discrepancies between the Notch2 and the DTR by Kaplan's approaches, which may mean that the remaining DC2 (lacking either Notch2 signaling or lacking Runx) could be the cells that mediate the impact on Ror γ t.

Minor points.

Is there a typo on page 6, "FpxP3 expression level was NOT decreased upon inactivation of cbfb in T cells" ??

What is point of Figure 3 and the role of the E-11 enhancer in this story. It has little to do with DC2 mechanism.

Reviewer #2 (Comments to the Authors (Required)):

In this paper, Tenno et al examine the role of Runx family transcription factors in the development and functions of intestinal dendritic cells. The results show that Runx1 and Runx3 play complementary roles in the maintenance of CD103⁺CD11b⁺ DC in the intestine and drive their ability to induce the generation of Ror γ ⁺ CD4⁺ T cells. CD11b⁺ cDC2 from mice lacking Runx in CD11c⁺ cells are transcriptionally distinct and these animals develop spontaneous colitis. Data are also provided to indicate that the Runx-binding enhancer E-11 in the Rorc gene is needed for the generation of Ror γ ⁺ T cells and ILC3. The findings on intestinal DC are novel, but are consistent with previous work in other models, including mice lacking the TGF β R in CD11c⁺ cells, a signalling pathway that involves Runx. These data are clear, but are mostly descriptive in nature and no mechanisms have been explored. A similar comment applies to data showing the failure of the Runx-deleted cDC2 to drive Ror γ ⁺ T cell differentiation, with the transcriptional findings that are presented being over-interpreted without direct evidence for the authors' conclusions. The experiments on the E-11 Runx binding enhancer in CD4⁺ T cells and ILC3 are not integrated clearly with the rest of the paper, as this is a distinct genetic element and the findings do not help interpretation of the work on deleting Runx itself in DC. Together with other, specific issues and sloppy presentation in places, these aspects reduce the impact of the work. Specific comments:

- 1) Despite the clear link between Runx and TGF β R signalling, the authors do not discuss how similar/different their findings are to those showing similar defects in CD103⁺CD11b⁺ intestinal DC in CD11c-cre-TGF β R mice. In fact it is important to note that the current results do not phenocopy the previous study, where it was shown that CD101 expression on total cDC2 was not altered in the absence of TGF β R signalling, suggesting that the defect in the cDC2 lineage was at a late stage of differentiation. This contrasts with the reduced CD101 expression found on total cDC2 in the current study, a finding which the authors interpret as Runx being of generic importance in cDC2 development. Data on cDC2 in other tissues would be important to assess this idea better.
- 2) There are some indications that Runx may play a role in CD103⁺CD11b⁻ cDC1 development, with for instance, a significant defect in this population amongst migratory DC in the MLN of CD11c-cre-Cbfb, Runx1 and Runx1/3 deficient mice. An effect on cDC1 would be consistent with findings from other models in which CD103⁺CD11b⁺ cDC2 are reduced, such as GM-CSF KO mice and warrants comment.
- 3) The fact that Runx seems to play a direct role in CD4⁺ T cell differentiation emphasises the need for examining the expression of the CD11c-cre driven Runx deletion in T cells. This is also necessary as the previous work in CD11c-cre-TGF β R mice showed off-target expression in T cells and this led to spontaneous colitis in these mice. As a result, DC had to be examined on a ragKO background and the effects of CD11c-cre-Runx deletion on T cell differentiation and colitis shown here cannot be interpreted accurately.
- 4) As noted above, the additional data examining the role of Runx and the E-11 enhancer binding element in T cells, thymocytes and ILC are not put in clear context with the studies of DC. In addition, these experiments are often not described clearly. For instance, the use of Rorc-GFP mice is not explained.
- 5) The data and references to the other mouse strains in which CD103⁺CD11b⁺ DC are defective

are over-interpreted. For instance, there is no specific evidence to support the authors' statements that Notch2 acts via IRF4 linked effects on survival, compared with an effect of Runx and TGFbR on development.

6) Consistent with a possible effect of Runx on CD103+CD11b- DC, the transcriptional studies shown in Figure 4 indicate substantial gene changes in these cells. However these data not discussed anywhere in the text.

7) It is not clear why IL18 was selected as the most important gene from the analysis of CD103-CD11b+ DC and no evidence is presented to substantiate the authors' conclusions that IL18 is linked to the cDC2 and Th17 defect in the Runx deficient mice.

8) As well as being difficult to interpret, the colitis studies are extremely limited, are not extended mechanistically and are of doubtful relevance.

Reviewer #3 (Comments to the Authors (Required)):

The manuscript by Tenno et al sets out to examine the function of Runx/Cbfb in gut conventional DC (cDC) and Rorgt+ T cells to address the wider question which cDC subset regulates the differentiation of Rorgt+ T cells and how these cells integrate signals from cDCs to activate Rorc gene expression. The paper addresses an important question in the field and is clearly written. Using genetic mouse models to delete Runx/Cbfb in cDCs and T cell populations, the authors report that Runx/Cbfb complexes are essential for the differentiation of gut CD103+CD11b+ cDC2s and that this results in impaired Rorgt+ T cell differentiation in the intestine. They also examine whether other cDC populations that are less obviously affected by loss of Runx/Cbfb may also contribute to the phenotype by changes at the molecular level. Gene expression analysis of Cbfb null CD103+CD11b- and CD103-CD11b+ cDCs show changes in CD103-CD11b+ cells that could affect the phenotype. Finally they assess the development of colitis in the Cbfb f/f:CD11c-Cre mice at 6 months of age and find they spontaneously develop colitis.

There are several important issues that need clarifying:

1) While the effect of loss of Cbfb on the CD103+11b+ cDCs is clear both in terms of frequency and cell numbers (Figure 1A, B), the cDC phenotype after Runx3 or Runx1 deletion seems very weak/non-existent in terms of absolute cell numbers. Only when both genes are deleted is an effect seen. This should be discussed and explained. For the mesenteric lymph node no absolute cell numbers are shown (EV1D); is this because they didn't show a change? In this tissue CD103+11b+ cDCs are not affected in relative frequency upon deletion of Runx1 or Runx3 alone. What is the explanation for this?

2) The expression of Runx factors was determined through data mining of existing RNA-Seq datasets. At the RNA level the expression of all 3 Runx factors is similar in the various cDC subset. Yet deletion of these factors individually has varying results. How can this be explained? It is important to assess expression of the Runx factors at the protein level by Ab staining of intestine sections. Particularly since it is well known that there can be large discrepancies between RNA and protein expression of Runx factors.

3) It is not clear which of the Runx factors is involved in the colitis, as this was examined only after deletion of Cbfb. The role of Runx/Cbfb complexes in gut biology has long been studied and there are conflicting reports in the literature. In other parts of the paper Runx1 and Runx3 floxed mice are analysed and analysis of colitis in these models would significantly add to the paper and be of interest to the field.

4) Why is colitis examined only at 6 months while it is known that comparable mouse models spontaneously develop colitis at 6 to 8 weeks? It could very well be that the Cbfb:CD11b-Cre mice also develop colitis much earlier. This needs to be examined as it would affect the remainder of the

study. If the mice develop colitis at a young adult age, all experiments were presumably done under inflammatory conditions (there is no mention of the age of the mice used in Figure 1 to 4). This is a major issue that would affect the interpretation of the data.

Other points:

5) There is no mention of the colon results in the text (Figure 2A)

6) Absolute cell numbers are only shown in Figure 1. For the remainder of the paper only relative phenotypic changes are shown. Absolute cell numbers should be provided.

7) Data is presented as the mean {plus minus} SD, but it is not clear what the individual data points represent. Do the dots in the graphs represent individual mice or pooled tissues from multiple mice, or individual experiments? There is no indication of the number of experiments performed. This is the case throughout the manuscript.

8) Care needs to be taken when comparing the effects of different mutations across separate experiments. Was this the case for the analysis of the Notch f/f:CD11c-Cre and Cbfb f/f :CD11c-Cre mice? Are the mice on the same genetic background?

9) It is not clear whether the RNA-seq was performed on Cbfb f/f:CD11c-Cre cDC populations or on straight Cbfb KO cells (Figure 4B, C). This is important to clarify as it can affect the result.

10) It is shown that expression of Rorc is dependent on a Runx-binding enhancer. To further corroborate a direct interaction and the conclusion that Runx factors play crucial roles in Th cells, the effect of RUNX motif mutation on enhancer activity/Rorc activity should be shown.

Point-by-point responses:

We thank three reviewers for their helpful comments, suggestions and constructive criticisms that have substantially improved the manuscript. One major criticism to the original manuscript was insufficient explanation of results and materials, and missing of citations in references. These were in part caused by the word limitation in a brief report, an article style in the original manuscript that we chose for initial submission and was transferred to Life Science Alliance. Taking advices by the reviewers and given opportunity for revision, we revised manuscript and changed the manuscript style to Research Article in Life Science Alliance to increase clarity of results and our points in discussion section.

Reviewer #1 (Comments to the Authors (Required)):

I am in favor of publication after minor changes to the discussion.

We thank the reviewer for positive evaluation on this work.

I agree with the summary provided by the authors at the end of the presentation of the results, which states "we provide concrete genetic evidence supporting a crucial function of Runx/Cbfbeta complexes in gut DC development not only for ... CD103+CD11b+ cDC2, but also for their polarizing activity that prime both Rorgt+ Th17 and type 3 Rorgt FoxP3+ Treg cells". This is a fair summary. And this summary is actually about all the paper really says. There are limitations in the study that are implicit in this statement. The evidence provided in the data is limited exclusively to genetic evidence, and there is absolutely no mechanistic data provided at any other level to explain the how loss of Cbfb in DCs causes a reduction in Rorgt expression in T cells. The genetic evidence applies only to the requirement for Runx/Cbfbeta in DCs. This data does not indicate that the decrease in Rorgt expression in T cells was even caused by the absence of the CD103+CD11b+ cDC2. Yes, that subset of DC2 is missing, but Runx/Cbfbeta complexes are missing from all types of other DCs that remain in the mouse. The genetic evidence does not indicate if the impact on Rorgt in T cells is due to loss of a DC population or from loss of function in the DCs that remain. Since the authors provide no functional mechanism for how the DCs of any type impact the development of the Rorgt+ T cells, the authors need to be very careful to not over extend their conclusions. They run the risk of adding to a dogma that CD103+ CD11b+ DC2 drive Rorgt in the T cells. No such evidence is there. This area in the DC field already has conflicting data as to these DC2 in the gut, with respect to various models, the deletion of Notch2 by CD11c-Cre compared to deletion of these cells by a transgene DT reagent generated by Kaplan used by several groups. The main point is that there is no

underlying mechanism for how any DC2 subset exerts a skewing on T cells (whether it is Notch2, Runx, KLF4). It is just not known at this time. The expression arrays shown in Figure 4 do not answer it.

We thank the reviewer for very thoughtful comments. We agree that our data do not formally exclude the possibility that loss of function of remaining cDCs by Runx/Cbfb deficiency also has impact on Ror γ t⁺ T cell development in the gut. We therefore modified abstract not to overextend our conclusion, and pointed out correlation of loss of CD103⁺CD11b⁺ cDC2 with loss of Rorgt⁺ T cells. We also discussed possibility that remaining cDCs subsets other than CD103⁺CD11b⁺ cDC2 in *Cbfb*^{FF}: *CD11c-Cre* mice have redundant functions (line 1-15, page12).

Regarding publication, I think that this journal is a fine venue to publish the data, as is, and I have no problems with any of the figures or experiments. I don't really like the inclusion of the E-11 enhancer data in this paper myself, but I can understand the need at times to combine orphaned data, and that is what this seems like to me. It adds a figure or two, but it really adds nothing related to the mechanism of the DC effect on T cells, in the end. But keep it in, and discuss separately.

We thank for positive comments for publication of this work in *Life Science Alliance* and supportive comments for keeping the *E-11* enhancer data in the manuscript. Although we agree that the *E-11* enhancer data is not directly related to the DCs function, we believe that this is informative data to understand T-cell intrinsic mechanisms that integrate DC-derived signals to activate the *Rorc* gene. However, since reviewers #1 and #2 found it difficult to connect the *E-11* enhancer data with DCs function, we moved these data into supplementary Figure S4 and discussed the *E-11* enhancer data separately in discussion section.

What I whole heartedly encourage the authors to do is to re-write much of the latter part and emphasize the parts of the story that still are totally unclear. By that I mean missing cells does not implicate them as the cause of the effect, as the Runx/Cbfb is gone in the other DCs as well, and some reviewers would even complain about the CD11c-Cre, since some deletion in other cells might occur, but I will not complain. The authors have under-cited some papers, and this journal's citation style is really annoying, I must say. The authors should also expose the discrepancies between the Notch2 and the DTR by Kaplan's approaches, which may mean that the remaining DC2 (lacking either Notch2 signaling or lacking Runx) could be the cells that mediate the impact on Rorgt.

We thank the reviewer for thoughtful and encouraging suggestions. We agree with the point that missing cells are not always cause for the effect. According to the reviewer's suggestion, we

re-wrote discussion with significant changes of manuscript style/structure, and discussed our data with published work including Langerin-DTR system (from line 3, page11).

Minor points.

Is there a typo on page 6, "FoxP3 expression level was NOT decreased upon inactivation of Cbfb in T cells" ?

We thank the reviewer for pointing out this typo. This was not typo. FoxP3 level per each cell was reduced by loss of Cbfb. For appropriate description, we add slightly in this sentence.

What is point of Figure 3 and the role of the E-11 enhancer in this story. It has little to do with DC2 mechanism.

This is the same point to above one regarding the *E-11* enhancer. Please see our reply there.

Reviewer #2 (Comments to the Authors (Required)):

In this paper, Tenno et al examine the role of Runx family transcription factors in the development and functions of intestinal dendritic cells. The results show that Runx1 and Runx3 play complementary roles in the maintenance of CD103b+CD11b+ DC in the intestine and drive their ability to induce the generation of Rorgt+ CD4+ T cells. CD11b+ cDC2 from mice lacking Runx in CD11c+ cells are transcriptionally distinct and these animals develop spontaneous colitis. Data are also provided to indicate that the Runx-binding enhancer E-11 in the Rorc gene is needed for the generation of Rorgt+ T cells and ILC3. The findings on intestinal DC are novel, but are consistent with previous work in other models, including mice lacking the TGFbR in CD11c+ cells, a signaling pathway that involves Runx. These data are clear, but are mostly descriptive in nature and no mechanisms have been explored. A similar comment applies to data showing the failure of the Runx-deleted cDC2 to drive Rorgt+ T cell differentiation, with the transcriptional findings that are presented being over-interpreted without direct evidence for the authors' conclusions. The experiments on the E-11 Runx binding enhancer in CD4+ T cells and ILC3 are not integrated clearly with the rest of the paper, as this is a distinct genetic element and the findings do not help interpretation of the work on deleting Runx itself in DC. Together with other, specific issues and sloppy presentation in places, these aspects reduce the impact of the work.

Specific comments:

1) Despite the clear link between Runx and TGFbR signaling, the authors do not discuss how similar/different their findings are to those showing similar defects in CD103+CD11b+ intestinal DC in

CD11c-cre-TGFbR mice. In fact, it is important to note that the current results do not phenocopy the previous study, where it was shown that CD101 expression on total cDC2 was not altered in the absence of TGFbR signaling, suggesting that the defect in the cDC2 lineage was at a late stage of differentiation. This contrasts with the reduced CD101 expression found on total cDC2 in the current study, a finding which the authors interpret as Runx being of generic importance in cDC2 development. Data on cDC2 in other tissues would be important to assess this idea better.

We thank the reviewer for his/her suggestion to analyze cDC2 in other tissues. We have analyzed DCs subsets in the lung of *Cbfb*^{+/+}: *CD11c-Cre* and *Cbfb*^{F/F}: *CD11c-Cre* mice and found that CD103⁻CD11b⁺ cDC2 subset was decreased in *Cbfb*^{F/F}: *CD11c-Cre* mice. Interestingly, CD101 expression was almost undetected on lung cDCs of *Cbfb*^{F/F}: *CD11c-Cre* mice. Thus, Runx/Cbfb complexes play important roles in regulating cDC2 differentiation in other barrier tissues. We show these lung data in supplementary Fig. S2B. We also added paragraph to discuss how gut cDC phenotypes differ between *Cbfb*^{F/F}: *CD11c-Cre* and *Tgfb1*^{F/F}: *CD11c-Cre* mice in the discussion section (line10-21, page 11).

2) There are some indications that Runx may play a role in CD103+CD11b- cDC1 development, with for instance, a significant defect in this population amongst migratory DC in the MLN of CD11c-cre-Cbfb, Runx1 and Runx1/3 deficient mice. An effect on cDC1 would be consistent with findings from other models in which CD103+CD11b+ cDC2 are reduced, such as GM-CSF KO mice and warrants comment.

We thank the reviewer for this excellent suggestion. In the revised manuscript, we added the graphs of the frequency and absolute cell numbers of cDCs subsets in large intestine and mesenteric LNs of *Runx1*, *Runx3*, and *Runx1/Runx3* double mutants in Fig. 2. As the reviewer pointed out, although the frequencies of CD103⁺CD11b⁻ cDC1 subset in mesenteric LNs tended to be reduced in *Runx1* mutant mice, absolute cell numbers of this subset were not significantly changed. Therefore, we hesitated to make a strong argument on the reduction of cDC1 subset by loss of Runx/Cbfb complexes.

3) The fact that Runx seems to play a direct role in CD4+ T cell differentiation emphasizes the need for examining the expression of the CD11c-cre driven Runx deletion in T cells. This is also necessary as the previous work in CD11c-cre-TGFbR mice showed off-target expression in T cells and this led to spontaneous colitis in these mice. As a result, DC had to be examined on a ragKO background and the effects of CD11c-cre-Runx deletion on T cell differentiation and colitis shown here cannot be interpreted accurately.

We thank the reviewer for pointing out a possible leaky expression of *CD11c-Cre* Tg in T cells. We examined Cre-mediated recombination at the *Cbfb* locus in gut CD4⁺ T cells by genomic DNA PCR and found that significant proportions of those cells underwent Cre-mediated recombination, which was shown in supplementary Fig. S1B in the revised manuscript. To compare gut T cell phenotypes caused by *Cbfb* inactivation by *CD11c-Cre* with those by *CD4-Cre*, we also examined *Cbfb^{F/F}; CD4-Cre* mice. As shown in Fig 3, although *Cbfb* inactivation in T cells by *CD4-Cre* resulted in an increase of Gata3⁺ Th cells with reduced level of FoxP3, reduction of Rorγt⁺ T cells, in particular Th17 cells, was not significant, compared to that in *Cbfb^{F/F}; CD11c-Cre* mice. This difference supports our conclusion that Runx/Cbfb function in cDCs is important to support Rorγt⁺ T cell differentiation. Furthermore, reduction of gut Rorγt⁺ T cells, in particular Rorγt⁺FoxP3⁺ Treg cells, become more severe by combinational *Cbfb* inactivation by *CD11c-Cre* and *CD4-Cre*, indicating that T-cell-intrinsic and Runx-dependent mechanisms that support differentiation of Rorγt⁺ T cells are present. To make these points clear, we modified text (from line 23, page 6, and line 12, page 7).

Contrary to spontaneous severe colitis development by loss of TGFβR in T cells by *CD4-Cre*, inactivation of *Cbfb* gene by *CD4-cre* did not result in spontaneous colitis development in 4 to 10 weeks-old young mice (data not shown), which is consistent with late onset of colitis in *Cbfb^{F/F}; CD11c-Cre* mice (Fig. 6).

4) As noted above, the additional data examining the role of Runx and the E-11 enhancer binding element in T cells, thymocytes and ILC are not put in clear context with the studies of DC. In addition, these experiments are often not described clearly. For instance, the use of *Rorc-GFP* mice is not explained.

We thank the reviewer for raising this point, which was also pointed out by reviewer#1. Please also see our reply to reviewer#1. We agree that the *E-11* enhancer data is not directly related to the cDCs function. However, we believe that distinct phenotypes in gut Rorγt⁺ T cell differentiation between *Cbfb^{F/F}; CD11c-Cre* and *Cbfb^{F/F}; CD4-Cre* and severe reduction of gut Rorγt⁺ T cell by combinational *Cbfb* inactivation by both *CD11c-Cre* and *CD4-Cre* indicate a presence of T-cell-intrinsic mechanisms that integrate DC-derived signals to activate the *Rorc* gene in a Runx-dependent manner. To our knowledges, enhancer(s) that is essential for *Rorc* activation during gut Th17 differentiation has not been reported. In addition, it is important to understand how cDCs-derived signals eventually activate Th17 program, which should be of interest also to researchers in DCs fields. Therefore, we believe that it is worth presenting identification and characterization of a single enhancer essential for Rorγt induction during

Th17 differentiation. We hope that reviewers #2 will find it significance to include our *E-11* enhancer data that is shown as supplementary figure 4 in the revised manuscript. To increase clarity of the *Rorc*-GFP reporter allele, we added more explanation (line 22, page 7).

5) The data and references to the other mouse strains in which CD103⁺CD11b⁺ DC are defective are over-interpreted. For instance, there is no specific evidence to support the authors' statements that Notch2 acts via IRF4 linked effects on survival, compared with an effect of Runx and TGFbR on development.

We thank reviewer#2 for such thoughtful criticisms. After careful reading of the reference manuscript (Persson et al., 2013), we agree that concrete evidence that links loss of Notch2 with IRF4 in impaired cDC survival was not shown. We therefore remove this statement.

6) Consistent with a possible effect of Runx on CD103⁺CD11b⁻ DC, the transcriptional studies shown in Figure 4 indicate substantial gene changes in these cells. However, these data not discussed anywhere in the text.

7) It is not clear why IL18 was selected as the most important gene from the analysis of CD103⁺CD11b⁺ DC and no evidence is presented to substantiate the authors' conclusions that IL18 is linked to the cDC2 and Th17 defect in the Runx deficient mice.

We thank reviewer#1 for pointing out insufficient explanation/discussion on transcriptome data. Accordingly, we added new paragraph on changes of gene signatures in CD103⁺CD11b⁻ cDC1 cells (line 18, page 9). During our literature search on putative functions of dysregulated genes in *Cbfb*-deficient cDCs, we noticed the paper proposing an inhibitory role of IL18 on Th17 differentiation and roles of IL22ra2 on intestinal inflammation. We agree that there is no evidence that IL18 or IL22ra2 is involved in impaired Th17 differentiation in *Cbfb*^{F/F}: *CD11c-Cre* mice. However, we believe that it is worth discussing IL18 upregulation and IL22ra2 down regulation from the views of possible mechanism that prevents Th17 differentiation and added some sentences in the discussion section (line 10, page 12).

8) As well as being difficult to interpret, the colitis studies are extremely limited, are not extended mechanistically and are of doubtful relevance.

We agree that our manuscript lacked mechanistic insights into colitis development. However, since *TGFbR*^{F/F}:*CD11c-cre* mice, which is related with *Cbfb*^{F/F}: *CD11c-Cre* mice, develop colitis and reviewer#3 requested to address effect of inflammation on differentiation of cDCs and T cells, we think that it is still worth showing gut histology data.

Reviewer #3 (Comments to the Authors (Required)):

The manuscript by Tenno et al sets out to examine the function of Runx/Cbfb in gut conventional DC (cDC) and Rorgt+ T cells to address the wider question which cDC subset regulates the differentiation of Rorgt+ T cells and how these cells integrate signals from cDCs to activate Rorc gene expression. The paper addresses an important question in the field and is clearly written. Using genetic mouse models to delete Runx/Cbfb in cDCs and T cell populations, the authors report that Runx/Cbfb complexes are essential for the differentiation of gut CD103+CD11b+ cDC2s and that this results in impaired Rorgt+ T cell differentiation in the intestine. They also examine whether other cDC populations that are less obviously affected by loss of Runx/Cbfb may also contribute to the phenotype by changes at the molecular level. Gene expression analysis of Cbfb null CD103+CD11b- and CD103-CD11b+ cDCs show changes in CD103-CD11b+ cells that could affect the phenotype. Finally, they assess the development of colitis in the Cbfb f/f:CD11c-Cre mice at 6 months of age and find they spontaneously develop colitis.

We thank reviewer#3 for positive evaluation on this work.

There are several important issues that need clarifying:

1) While the effect of loss of Cbfb on the CD103+11b+ cDCs is clear both in terms of frequency and cell numbers (Figure 1A, B), the cDC phenotype after Runx3 or Runx1 deletion seems very weak/non-existent in terms of absolute cell numbers. Only when both genes are deleted is an effect seen. This should be discussed and explained. For the mesenteric lymph node, no absolute cell numbers are shown (EVID); is this because they didn't show a change? In this tissue CD103+11b+ cDCs are not affected in relative frequency upon deletion of Runx1 or Runx3 alone. What is the explanation for this?

We thank the reviewer for pointing out insufficient presentation of data. We added frequency and cell numbers of small intestine, large intestine and mesenteric LN of Cbfb mutant (Fig. 1 and supplementary Fig. S1) and Runx1/3 mutant mice (Fig. 2) in the revised manuscript. As for the functional redundancy between Runx1 and Runx3, we have shown that both proteins can compensate each other for silencing Cd4 gene and mature thymocyte generation (Setoguchi R. Science 319:816, 2018). Thus, redundant function of Runx1 and Runx3 during T cell development is well established concept. Therefore, it is natural that Runx1 and Runx3 have redundant function during differentiation of CD103+CD11b+ cDC2. We describe this point at line 4, page 6.

2) The expression of Runx factors was determined through data mining of existing RNA-Seq datasets. At the RNA level the expression of all 3 Runx factors is similar in the various cDC subset. Yet deletion of these factors individually has varying results. How can this be explained? It is important to assess expression of the Runx factors at the protein level by Ab staining of intestine sections. Particularly since it is well known that there can be large discrepancies between RNA and protein expression of Runx factors.

We thank the reviewer for this thoughtful suggestion. Although we tried to stain Runx proteins on intestinal sections, it was impossible to dissect three DCs subsets (CD103⁺CD11b⁻, CD103⁺CD11b⁺ and CD103⁻CD11b⁺) by immune histochemical analyses. We therefore examined Runx protein expression by flow-cytometer analyses. We were fortunate to get good antibody that specifically reacts and detects Runx3 expression. We found that Runx3 expression level is higher in CD11b⁺ DC subsets and showed this new result in supplementary Fig. S3B in the revised manuscript. Unfortunately, there were no such antibodies available for Runx1 and Runx2 expression.

3) It is not clear which of the Runx factors is involved in the colitis, as this was examined only after deletion of Cbfb. The role of Runx/Cbfb complexes in gut biology has long been studied and there are conflicting reports in the literature. In other parts of the paper Runx1 and Runx3 floxed mice are analyzed and analysis of colitis in these models would significantly add to the paper and be of interest to the field.

We thank reviewer#3 for raising this point. In terms of gut cDCs development, severe reduction of CD103⁺CD11b⁺ cDC2 was observed only when both Runx1 and Runx3 genes were inactivated, although single *Runx1* or *Runx3* inactivation tended to cause reduction of these cells. These observations indicate that Runx1 and Runx3 have redundant function to support cDC2 development. Therefore, it is difficult to conclude which Runx proteins is dominantly involved in cDC development from our data. In terms of colitis development, Runx1 and Runx3 double mutant mice (*Runx1/3^{F/F}:CD11c-Cre*) tended to develop colitis after 6 months as was observed in *Cbfb^{F/F}:CD11c-Cre* mice. However, since numbers of these mice were limited (just two mice), we hesitated to include this preliminary results in the revised manuscript.

4) Why is colitis examined only at 6 months while it is known that comparable mouse models spontaneously develop colitis at 6 to 8 weeks? It could very well be that the Cbfb:CD11b-Cre mice also develop colitis much earlier. This needs to be examined as it would affect the remainder of the study. If the

mice develop colitis at a young adult age, all experiments were presumably done under inflammatory conditions (there is no mention of the age of the mice used in Figure 1 to 4). This is a major issue that would affect the interpretation of the data.

We thank reviewer#3 for raising this question. We added gut histological results at 7 weeks-old and 4 months-old (Fig. 6 and supplementary Fig. S5) in the revised manuscript. At the 6 to 7 weeks old mice when we examined differentiation of gut cDCs and T cells by flow cytometry analyses, there were no colitis development. Therefore, it is unlikely that impaired differentiation of gut cDC and T cell subsets in *Cbfb^{F/F}:CD11c-Cre* mice is caused by inflammation. We agree that there are reports showing colitis development in young Runx3 germline KO mice and in recipient mice that received Runx3-deficient bone marrow transplantation. In such settings, Runx3 expression is lost in all hematopoietic cells. This could accelerate colitis development. In our current study, we focused on Runx/Cbfb function in cDCs, and found that loss of Cbfb in cDCs alone is not enough for colitis development at younger mice. These are our results that would be of interest to researchers in Runx field.

Other points:

5) There is no mention of the colon results in the text (Figure 2A).

We added sentence to describe colon results (line 21, page 4)

6) Absolute cell numbers are only shown in Figure 1. For the remainder of the paper only relative phenotypic changes are shown. Absolute cell numbers should be provided.

We added the graphs of absolute cell numbers of larger intestine and mesenteric LNs (Fig.1 and supplementary Fig. S1).

7) Data is presented as the mean {plus minus} SD, but it is not clear what the individual data points represent. Do the dots in the graphs represent individual mice or pooled tissues from multiple mice, or individual experiments? There is no indication of the number of experiments performed. This is the case throughout the manuscript.

We thank reviewer#3 for this suggestion. We corrected the explanations of graphs in the legends. Each dot represents individual mouse.

8) Care needs to be taken when comparing the effects of different mutations across separate experiments.

Was this the case for the analysis of the Notch f/f:CD11c-Cre and Cbfb f/f :CD11c-Cre mice? Are the mice on the same genetic background?

We thank reviewer#3 for this suggestion. Both *Notch2^{F/F}:CD11c-Cre* and *Cbfb^{F/F}:CD11c-Cre* were mixed genetic background of C57/B6 and 129.

9) It is not clear whether the RNA-seq was performed on Cbfb f/f:CD11c-Cre cDC populations or on straight Cbfb KO cells (Figure 4B, C). This is important to clarify as it can affect the result.

We thank reviewer#3 for this suggestion. We added the explanations into the text and figure legend. Actually, we used cDC population from *Cbfb^{F/F}:CD11c-Cre* mice for RNA-seq analysis.

10) It is shown that expression of Rorc is dependent on a Runx-binding enhancer. To further corroborate a direct interaction and the conclusion that Runx factors play crucial roles in Th cells, the effect of RUNX motif mutation on enhancer activity/Rorc activity should be shown.

We thank reviewer#3 for this suggestion. We fully agree that targeting specific mutation onto two Runx motifs within the *E-11* enhancer is a nice experiment to further confirm requirement of Runx binding for activating this enhancer. However, as reviewer#1 and #2 pointed out, characterization of this enhancer is not a major aim in this study. And generating and characterization of another two or three gene-edited mice will takes more than six months. We hope that reviewer#3 agree that including results of Runx motif mutations is beyond the scope of this work and is not definitive necessary for publication of this work in a timely manner.

November 21, 2019

RE: Life Science Alliance Manuscript #LSA-2019-00441-TR

Dr. Ichiro Taniuchi
RIKEN, Center for Integrative Medical Sciences (IMS)
RIKEN Center for Integrative Medical Sciences
1-7-22, Suehiro-cho, Turumi-ku
Yokohama, Kanagawa 230-0045
Japan

Dear Dr. Taniuchi,

Thank you for submitting your revised manuscript entitled "Essential functions of Runx/Cbfb in gut conventional dendritic cells for priming Ror γ t⁺ T cells". As you will see, the reviewers appreciate the changes introduced in revision and are now more supportive of publication, pending some final minor revisions. We would thus like to ask you to submit a final version to us:

- Please address the remaining concerns of the two reviewers
- Please make sure that the author order in manuscript and submission system is the same
- Even though clear from your material and methods section, please additionally mention in the figure legend next to the p-values which statistical test has been used
- Please add panel descriptors "A" and "B" to Fig S5 (note that this figure is currently labeled as figure S6)
- It would be good to change the color of the scale bars in Fig 6A and S5

A. FINAL FILES:

B. MANUSCRIPT ORGANIZATION AND FORMATTING:

Sincerely,

Reviewer #2 (Comments to the Authors (Required)):

I thank the authors for their detailed replies to my original comments and for providing additional information. As a result, the manuscript has been much improved and only one or two issues remain to be addressed:

- 1) The new data on the effects of deleting *Cbfb* in lung DC are interesting. However they do tend to confirm my initial impression that there may be a partial loss of CD103+CD11b- DC in these mice. While I appreciate the authors' caution in not wishing to over-interpret these findings, they do warrant comment on the grounds of the marked transcriptional differences seen in these cells in the intestine of the KO mice, the abolition of CD101 expression under these circumstances and the previous literature I referred to previously.
- 2) Similarly, the authors still do not comment appropriately on the transcriptional changes seen in CD103+CD11b- DC in the absence of *Cbfb*.
- 3) The marked effect of deleting *Cbfb* or *Runx1/3* on CD101 expression is interesting and as CD101 is believed to be driven by retinoic acid signalling, it would be interesting to know if there is any overlap between these pathways.

Reviewer #3 (Comments to the Authors (Required)):

The authors have addressed most of my concerns. It is not clear however why CD103+CD11b-, CD103+CD11b+ and CD103-CD11b+ could not be discriminated and *Runx3* expression shown by immunohistology: they seem to separate clearly enough in the flowcytometry plots (fig S3B).

Point-by-point responses:

We thank the reviewers for evaluating our revised manuscript. We are glad to hear their positive comments on our revision.

Reviewer #2 (Comments to the Authors (Required):

I thank the authors for their detailed replies to my original comments and for providing additional information. As a result, the manuscript has been much improved and only one or two issues remain to be addressed:

We are glad to hear that reviewer#2 considered that our revision significantly improved the manuscript.

1) The new data on the effects of deleting *Cbfb* in lung DC are interesting. However they do tend to confirm my initial impression that there may be a partial loss of CD103⁺CD11b⁻ DC in these mice. While I appreciate the authors' caution in not wishing to over-interpret these findings, they do warrant comment on the grounds of the marked transcriptional differences seen in these cells in the intestine of the KO mice, the abolition of CD101 expression under these circumstances and the previous literature I referred to previously.

As the reviewer noticed, CD103⁺CD11b⁻ DC subset tended to be reduced by loss of *Cbfb*. However, our statistical analysis did not show significance, letting us to hesitate to make strong argument on this point. We hope to publish this aspect in the future and deeper analyses of CD103⁺CD11b⁻ DC various tissues requiring more replicates.

On the other hand, we discussed significant changes in gene expression signatures of CD103⁺CD11b⁻ DC subset (page 11). According to the reviewer's suggestion, we added sentences to discuss differences in CD101 down-regulation between *Cbfb* and *Tgfb1* cKO and pointed out the possibility that loss of Runx/Cbfb affects DC development from earlier stage rather than loss of TGFβ signaling.

2) Similarly, the authors still do not comment appropriately on the transcriptional changes seen in CD103⁺CD11b⁻ DC in the absence of *Cbfb*.

We thank the reviewer for his/her suggestion. However, we are afraid that the reviewer might be misled, since we described transcriptional changes observed in CD103⁺CD11b⁻ DC subset in the revised manuscript. We also highlighted down-regulation of *IL22ra2* gene in both CD103⁺CD11b⁻ and CD103⁻CD11b⁺ DC subsets (page 9) and discussed possible involvement of IL18 in *IL22ra2* downregulation as well as possible functional changes of CD103⁺CD11b⁻

subset (page 12). We hope that the reviewer#2 agree that these are appropriate discussion points within the limited word counts.

3) The marked effect of deleting Cbfb or Runx1/3 on CD101 expression is interesting and as CD101 is believed to be driven by retinoic acid signalling, it would be interesting to know if there is any overlap between these pathways.

We thank the reviewer for this suggestion. We agree that it is interesting to examine overlap pathways between Runx and retinoic acid (RA) signaling. There are some examples of cell types whose differentiation is regulated by both Runx and RA, such as gut Treg cells. Synergic effect of RA on TGF β signaling is well established during Treg differentiation, and Runx proteins have been shown to interact with Smads molecules, known as major signal transducer of TGF β signaling. Thus, it is possible that Runx and RA pathways are overlapped in regulating CD101 expression as well as cDC2 development. However, we do not have direct data showing molecular interactions and intersections of these two pathways in above biological processes. Hence we believe it is safer to describe such possibility in another manuscript with better supporting experimental results.

Reviewer #3 (Comments to the Authors (Required)):

The authors have addressed most of my concerns.

We are glad to hear that reviewer#3 found that our revisions improved our manuscript.

It is not clear however why CD103⁺CD11b⁻, CD103⁺CD11b⁺ and CD103⁻CD11b⁺ could not be discriminated and Runx3 expression shown by immunohistology: they seem to separate clearly enough in the flowcytometry plots (fig S3B).

Difficulty in the discrimination of gut DC subsets by immunohistochemistry is largely due to technical issues. During our trials for immunohistochemistry, we noticed that combinational staining with antibodies for MHC-II, CD11c, CD1103 and CD11b with different colors is very difficult. In particular, discrimination of three CD103⁺CD11b⁻, CD103⁺CD11b⁺ and CD103⁻CD11b⁺ subsets requires quantitative analyses of CD103 and CD11b expression level, which are difficult in immunohistochemical analyses. We therefore had to use flow-cytometer analyses to examine Runx protein expression in these gut DC subsets.

November 27, 2019

RE: Life Science Alliance Manuscript #LSA-2019-00441-TRR

Dr. Ichiro Taniuchi
RIKEN Center for Integrative Medical Sciences
1-7-22, Suehiro-cho, Turumi-ku
Yokohama, Kanagawa 230-0045
Japan

Dear Dr. Taniuchi,

Thank you for submitting your Research Article entitled "Essential functions of Runx/Cbfb in gut conventional dendritic cells for priming Rorgt+ T cells.". I appreciate the introduced changes and it is a pleasure to let you know that your manuscript is now accepted for publication in Life Science Alliance. Congratulations on this interesting work.

*****IMPORTANT:** If you will be unreachable at any time, please provide us with the email address of an alternate author. Failure to respond to routine queries may lead to unavoidable delays in publication.*******

DISTRIBUTION OF MATERIALS:

Again, congratulations on a very nice paper. I hope you found the review process to be constructive and are pleased with how the manuscript was handled editorially. We look forward to future exciting submissions from your lab.

Sincerely,
